# Seroprevalence of brucellosis and molecular characterization of *Brucella* spp. from slaughtered cattle in Rwanda

Jean Bosco Ntivuguruzwa[1,2☯]*, Francis Babaman Kolo[1‡], Emil Ivan Mwikarago[3¤‡], Henriette van Heerden[1☯]

**1** Department of Veterinary Tropical Diseases, Faculty of Veterinary Science, University of Pretoria, Pretoria, South Africa, **2** Department of Veterinary Medicine, College of Veterinary Medicine, University of Rwanda, Kigali, Rwanda, **3** Department of Biomedical Services, National Reference Laboratory Division, Rwanda Biomedical Centre, Kigali, Rwanda

☯ These authors contributed equally to this work.
¤ Current address: Department of Laboratory Sciences, College of Medicine and Health Sciences, University of Rwanda, Kigali, Rwanda
‡ FBK and EIM also contributed equally to this work.
* boscus2@gmail.com

**Data Availability Statement:** All relevant data are within the paper and its Supporting Information files.

## Abstract

Bovine brucellosis is endemic in Rwanda, although, there is a paucity of documented evidence about the disease in slaughtered cattle. A cross-sectional study was conducted in slaughtered cattle (n = 300) to determine the seroprevalence of anti-*Brucella* antibodies using the Rose Bengal Test (RBT), and indirect enzyme-linked immunosorbent assay (i-ELISA). Corresponding tissues were cultured onto a modified Centro de Investigación y Tecnología Agroalimentaria (CITA) selective medium and analysed for *Brucella* spp. using the 16S-23S ribosomal interspacer region (ITS), AMOS, and Bruce-ladder PCR assays. The seroprevalence was 20.7% (62/300) with RBT, 2.9% (8/300) with i-ELISA, and 2.9% (8/300) using both tests in series. *Brucella*-specific 16S-23S ribosomal DNA interspace region (ITS) PCR detected *Brucella* DNA in 5.6% (17/300; *Brucella* culture prevalence). AMOS-PCR assay identified mixed *B. abortus* and *B. melitensis* (n = 3), *B. abortus* (n = 3) and *B. melitensis* (n = 5) while Bruce-ladder PCR also identified *B. abortus* (n = 5) and *B. melitensis* (n = 6). The gold standard culture method combined with PCR confirmation identified 5.6% *Brucella* cultures and this culture prevalence is higher than the more sensitive seroprevalence of 2.9%. This emphasizes the need to validate the serological tests in Rwanda. The mixed infection caused by *B. abortus* and *B. melitensis* in slaughtered cattle indicates cross-infection and poses a risk of exposure potential to abattoir workers. It is essential to urgently strengthen a coordinated national bovine brucellosis vaccination and initiate a test-and-slaughter program that is not presently applicable in Rwanda.

## Introduction

Brucellosis is a contagious widespread disease that causes not only substantial economic losses related to abortions, long conception intervals, and sterility in animals but also morbidity and

**Funding:** This study was supported by the Belgian Directorate-General for Development Cooperation, through its Framework Agreement with the Institute of Tropical Medicine (FA DGD-ITM 2017 – 2021). The funders had no role in study design, data collection and analysis, decision to publish, or preparation of the manuscript.

**Competing interests:** The authors have declared that no competing interests exist.

reduced working capacity in humans [1, 2]. The disease is caused by bacteria of the genus *Brucella* which belongs to the family of alphaproteobacteria [3]. *Brucella* species are gram-negative microaerophilic coccobacilli, acid-fast intracellular, and host-specific microorganisms affecting a wide variety of terrestrial and marine mammals [4, 5]. *Brucella* species are 96% genetically identical [6] with few polymorphisms that are essential for species and biovars differentiation [7, 8]. Classical species with their biovars (bv.) have specific hosts, for instance, *B. abortus* (7 biovars) infect primarily cattle, *B. melitensis* (3 biovars) infect goats and sheep, *B. ovis* infects sheep, *B. suis* (bv. 1, 3, 4, and 5) infect swine while *B. suis* bv. 2 infects rats, *B. canis* infects dogs, and *B. neotomae* infects wood rats [4, 9].

The transmission of brucellosis in animals is through inhalation of *Brucella* aerosols [10], direct contact with infective fetal membranes, vaginal discharges, placenta content, and ingestion of contaminated feeds [11]. There are no pathognomonic clinical signs for brucellosis, but cases of abortion or hygroma are suspicious signs that require laboratory diagnosis for confirmation [9, 12].

To detect brucellosis at the herd level, the most suitable tests are serological tests to determine the seroprevalence of brucellosis in the animal and or herd using a screening agglutination test, the Rose Bengal Test (RBT), and a confirmatory test like enzyme-linked immunosorbent assays (ELISAs) or complement fixation test (CFT) [9]. However, serological tests do not provide a complete diagnosis, thus, the isolation of *Brucella* spp. remains the gold standard [9]. The culturing and biotyping of *Brucella* cultures are expensive, time-consuming, and require trained personnel [7]. PCR assays differentiate *B. abortus* bv.1, 2, 4, *B. melitensis* bv.1, 2, 3, *B. ovis*, and *B. suis* bv.1 (AMOS PCR) in 24 hours from cultures [7] and Bruce-ladder PCR assay can differentiate all *Brucella* species and vaccine strains [13, 14]. Unfortunately, culture, phenotypic and genotypic isolation of *Brucella* spp. are not common in veterinary services in most developing countries owing to inadequate facilities and trained personnel; therefore, serology is in common practice with little knowledge on the type of infecting *Brucella* spp. [15].

Brucellosis is an endemic disease in Rwanda with a reported 7.4% to 18.7% seroprevalence in cattle [16, 17], as well as seroprevalence in women with a history of abortions varying between 6.1% and 25.0% [18, 19]. Although cattle from various districts of the country are slaughtered at abattoirs, there is no single study on the seroprevalence of brucellosis in slaughtered cattle in Rwanda. Furthermore, apart from a single study that isolated *B. abortus* bv. 3 from Rwandan cattle in the 1980s [20], *Brucella* spp. that are circulating in Rwanda are not known. The objective of this study was, therefore, to determine the seroprevalence of brucellosis and characterize *Brucella* spp. from slaughtered cattle in Rwanda. Our findings are essential to building an epidemiological database essential for the control of brucellosis in Rwanda.

## Materials and methods

### Study area

This study was conducted at six abattoirs in Rwanda. Rwanda is a landlocked country of the East African community covering an area of 26,338 Km$^2$ in the southern hemisphere near the equator (West: 28.86; East: 30.89; North: - 1.04; South: - 2.83). The bovine population in Rwanda was estimated at 1,293,768 in 2018 [21]. The six abattoirs (société des abattoirs de Nyabugogo "SABAN", Rugano abattoir, Kamembe, Rubavu, Kamuhanda, Gataraga) consented to participate (Fig 1). These abattoirs were selected based on their slaughtering capacity and their location to sample cattle from all thirty districts of Rwanda. In this study, cattle sampled at the SABAN abattoir were from 19 districts including Rulindo, Ngoma, Muhanga, Nyagatare, Gasabo, Bugesera, Ngororero, Gakenke, Burera, Rutsiro, Gicumbi, Nyarugenge,

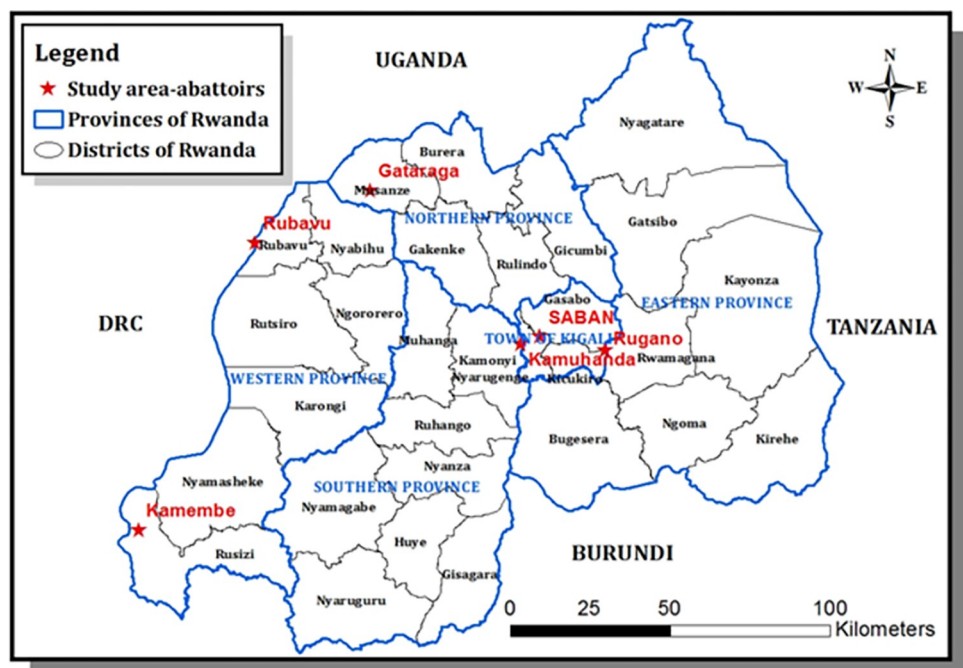

**Fig 1. A map of Rwanda with provinces and districts with red stars showing the locations of abattoirs visited in this study** [22].

Kirehe, Ruhango, Kayonza, Karongi, Nyanza, Kamonyi, and Gatsibo. Cattle sampled at Rugano abattoir were from three districts including Gasabo, Rwamagana, and Nyarugenge. Cattle sampled at Kamembe abattoir were from eight districts including Gisagara, Huye, Nyamagabe, Nyamasheke, Nyanza, Nyaruguru, Ruhango, and Rusizi. Cattle sampled at Rubavu abattoir were from two districts Nyabihu, and Rubavu. Cattle sampled at Kamuhanda abattoir were from the Kamonyi district. Cattle sampled at Gataraga abattoir were from the Musanze district. These abattoirs were classified into high throughput abattoirs (n = 4) slaughtering more than 50 cattle daily and low throughput abattoirs (n = 2) slaughtering 50 or less every day.

## Study design and sample size

A cross-sectional study was carried out from August 2018 through October 2019 to determine the seroprevalence of brucellosis and characterize *Brucella* spp. from cattle tissue selected during slaughtering at abattoirs. The sample size was calculated using the previously described formula [23] for cross-sectional studies.

$$N = \frac{Z^2\ P(1-P)}{d^2}$$

Where N is the sample size, $Z^2$ = 1.96 the statistical constant at a 95% confidence interval; P is the expected prevalence and was 0.5% based on previous studies [17], and the absolute precision, d = (P/2). According to the formula, the total sample size was 291 but it was rounded to 300 cattle to sample ten animals per each of the 30 districts of Rwanda.

## Sampling procedure

Our target was to sample five animals coming from the same district every day. The district of origin of animals was recorded on arrival using the movement permit issued by the sector

animal resources officer at the animal market. The age was determined using teeth erosion as previously described [24]. Except for abattoirs that received mostly males, females of one year and above were selected using systematic random sampling. Animals were aligned in a crush and every fourth animal was selected for sampling. The vaccination status and farm of origin of slaughtered animals could not be traced because most of the animals were bought from different animal markets.

## Collection of blood and tissues samples

After the selection and recording of individual demographic information (district of origin, age, breed, and sex), the animal was restrained, marked on the head, and released for resting waiting for the collection of blood after bleeding. Blood was collected into sterile 50 ml tubes after slaughter, aliquoted into 5 ml tubes, and immediately transported to the laboratory of the University of Rwanda (UR) and left overnight at room temperature to allow clotting. The following day, serum was collected into a sterile 2 ml micro-centrifuge tube and stored at -20°C until serological testing at Rwanda Agriculture and Animal Resources Board (RAB), Department of Veterinary Services, in the serology section. The head of the marked animal from which blood was collected was followed at the head inspection station and the corresponding left and right retropharyngeal lymph nodes were collected into a sterile 50 ml tube.

## Serological tests

Animal sera were tested with the RBT following the manufacturer's guidelines (IDvet, France) and the OIE protocol [9]. Briefly, equal volumes (30 μl) of *Brucella* antigens and sera were gently mixed for four minutes, and any agglutination was regarded as a positive result. All sera were also checked for the presence of anti-*Brucella* antibodies with a confirmatory test kit namely a multispecies i-ELISA according to manufacturer's guidelines (IDvet, France) and the OIE protocol [9] with positive and negative controls. The cut-off point for the seropositivity was 120% and sera samples having 120% optical densities were considered positive. These serological tests were chosen because of their combined effects of sensitivity and specificity [25]. RBT is a screening test with high sensitivity, while i-ELISA is a confirmatory test with high specificity [25, 26]. Any detection of anti-*Brucella* antibodies by RBT or i-ELISA was considered to determine the seroprevalence of brucellosis.

## Culturing and *Brucella* isolation from tissues

Tissue samples were processed and cultured in a biosafety level 3 (BSL 3) facility at the National Reference Laboratory (NRL), Rwanda Biomedical Centre (RBC), Kigali, Rwanda according to the guidelines previously described [9]. Briefly, tissues were sliced using sterile scissors and forceps into sterile mortars and grounded using a sterile pestle. An aliquot of pooled homogenate was spread into a modified Centro de Investigación y Tecnología Agroalimentaria (CITA) selective medium and incubated at 37°C with a 10.0% $CO_2$ atmosphere [27]. Plates were read for bacterial growth every day for four weeks. The DNA was extracted from colonies suspected of *Brucella* organisms.

## DNA extraction and identification of the genus *Brucella* spp.

Genomic DNA was extracted from cultures using the ReliaPrep gDNA tissue Miniprep system following the manufacturer's guidelines (Promega, USA). This DNA was screened for *Brucella* DNA using *Brucella*-specific primers (Table 1) designed from a gene-specific 16S -23S rDNA interspacer region (ITS) [28] with the *B. abortus* RF 544 (Onderstepoort Biological Products,

**Table 1. Sequences of oligonucleotide primers used for the distinction of *Brucella* species isolated from slaughtered cattle in Rwanda.**

| PCR name | Primer name | Sequence (5'-3') | Targets | Size (bp) | Conc. (µM) | References |
|---|---|---|---|---|---|---|
| ITS | ITS66 f | ACATAGATCGCAGGCCAGTCA | *16s-23s rDNA* | 214 | 0.2 | [28] |
| | ITS279 r | ACATAGATCGCAGGCCAGTCA | | | | |
| AMOS | *B. abortus* | GAC GAA CGG AAT TTT TCC AAT CCC | IS711 | 498 | 0.1 | [7] |
| | *B. melitensis* | AAA TCG CGT CCT TGC TGG TCT GA | | 731 | 0.1 | |
| | *B. ovis* | CGG GTT CTG GCA CCA TCG TCG GG | | 976 | 0.1 | |
| | *B. suis* | GCG CGG TTT TCT GAA GGT GGT TCA | | 285 | 0.1 | |
| | *IS 711* | TGC CGA TCA CTT AAG GGC CTT CAT | | | 0.2 | |
| BRUCE- LADDER | BMEI0998f | ATC CTA TTG CCC CGA TAA GG | *wboA* | 1682 | 6.25 | [29, 30] |
| | BMEI0997r | GCT TCG CAT TTT CAC TGT AGC | | | | |
| | BMEI0535f | GCG CAT CT TCG GTT ATG AA | *bp26* | 450 | 6.25 | [31] |
| | BMEI0536r | CGC AGG CGA AAA CAG CTA TAA | | | | |
| | BMEII0843f | TTT ACA CAG GCA ATC CAG CA | *omp31* | 1071 | 6.25 | [32] |
| | BMEII0844r | GCG TCC AGT TGT TGT TGA TG | | | | |
| | BMEI1436f | ACG CAG ACG ACC TTC GGT AT | *Deacetylase* | 794 | 6.25 | [33] |
| | BMEI1435r | TTT ATC CAT CGC CCT GTC AC | | | | |
| | BMEII0428f | GCC GCT ATT ATG TGG ACT GG | *eryC* | 587 | 6.25 | [34] |
| | BMEII0428r | AAT GAC TTC ACG GTC GTTCG | | | | |
| | BR0953f | GGA ACA CTA CGC CAC CTT GT | *ABC Transporter* | 272 | 6.25 | [35] |
| | BR0953r | GAT GGA GCA AAC GCT GAA G | | | | |
| | BMEI0752f | CAG GCA AAC CCT CAG AAG C | *rpsL* | 218 | 6.25 | [36] |
| | BMEI0752r | GAT GTG GTA ACG CAC ACC AA | | | | |
| | BMEII0987f | CGC AGA CAG TGA CCA TCA AA | *CRP Regulator* | 152 | 6.25 | [33] |
| | BMEII0987r | GTA TTC AGC CCC CGT TAC CT | | | | |

South Africa) as a positive control. For the negative control, we used ultrapure sterile water (Onderstepoort Biological Products, South Africa). The 15 µl PCR reaction mixture contained 1x of MyTaq^TM Red PCR Mix (Bioline, Johannesburg, South Africa), primers at 0.2 µM, and 2 µl of template DNA. The PCR cycling condition was initial denaturation at 95˚C for 3 min followed by 35 cycles of denaturation at 95˚C for 1 min, annealing at 60˚C for 2 min, and extension at 72˚C for 2 min and a final extension step at 72˚C for 5 min. The primers amplified a 214 bp fragment that was analyzed by electrophoresis using a 2% agarose gel stained with red gel nucleic acid stain and visualized under UV light. Molecular analyses were done at the Department of Veterinary Services, Rwanda Agriculture Board (RAB) Kigali, Rwanda.

## Identification of *Brucella* species using AMOS and Bruce-ladder PCR assays

The DNA samples that were ITS PCR positive were tested for *B. abortus*, *B. melitensis*, *B. ovis*, and *B. suis* using a multiplex AMOS PCR assay as previously described [7]. A 25 µl reaction mixture contained 1x MyTaq Red PCR Mix (Bioline, Johannesburg, South Africa), four species-specific forward primers and reverse primer IS711 (Table 1) at a final concentration of 0.1 µM and 0.5 µM respectively, and 2 µl of template DNA. Thermocycling conditions included initial denaturation at 95˚C for 3 min followed by 35 cycles of denaturation at 95˚C for 1 min, annealing at 60˚C for 2 min, an initial extension at 72˚C for 2 min, and a final extension at 72˚C for 5 min. PCR products were analysed by gel electrophoresis using 2% agarose stained with red gel nucleic acid stain and visualized under UV light.

Vaccine strains and field isolates of *Brucella* spp. were identified and differentiated by a multiplex Bruce-ladder PCR as previously described [14]. A 25 μl PCR reaction contained 1x MyTaq™ Red Mix (Bioline, Johannesburg, South Africa), eight species-specific forward and reverse primers at a final concentration of 6.25 μM (Table 1), and 5 μl of template DNA. The PCR cycling conditions included an initial denaturation at 95˚C for 5 min followed by 25 cycles at 95˚C for 30 s, at 64˚C for 45 s, and at 72˚C for 3 min, and a final extension step at 72˚C for 10 min. The expected product sizes in AMOS PCR assay are 498 bp and 731 bp for *B. abortus* and *B. melitensis*, respectively. The expected product sizes for *B. abortus* in Bruce-ladder PCR assay are 152 bp, 498 bp, 587 bp, 587 bp, and 794 bp; they are 152 bp, 498 bp, 587 bp, 794 bp, and 1071 bp for *B. melitensis*. PCR products were analysed by gel electrophoresis using a 2% agarose stained with gel red nucleic acid stain and viewed under UV light.

## Data analysis

The overall seroprevalence was obtained by dividing the total number of animals simultaneously positive to RBT and i-ELISA by the total number of animals sampled. Data were recorded in Microsoft Excel spreadsheets. Epi-Info 7 version 10 was used to calculate proportions. Significant levels between individual risk factors and results from cultures confirmed by ITS PCR assay were determined using the chi-square test. The odds ratios were determined for associated risk factors along 95% confidence intervals and statistical significance set at $p < 0.05$. The brucellosis case status was defined based on the results of cultures and the ITS PCR test in series. The chi-square test was used to evaluate the univariable association between the explanatory variables and brucellosis case status. Any explanatory associated with brucellosis at a $P \leq 0.20$ was included in the multivariable logistic regression analysis to identify risk factors.

## Ethics statement

The authorization to conduct the study was obtained from the research screening and ethical clearance committee of the College of Agriculture, Animal Sciences and Veterinary Medicine, University of Rwanda (Ref: 026/DRIPGS/2017), institutional review board of the College of Medicine and Health Sciences, University of Rwanda (N˚ 006/CMHS IRB/2020), and Animal Ethics Committee of the Faculty of Veterinary Science, University of Pretoria, South Africa (V004/2020). Informed verbal consent was obtained from managers of abattoirs, and owners of animals at the abattoirs.

## Results

### Descriptive statistics

Of the 300 cattle sera, 95.7% (287/300) were from females, while 4.3% (13/300) were from males. Most animals, 89.7% (269/300) were adults while young animals represented 10.3% (31/300). Twenty-seven percent [27.7%, (81/300)] of cattle sampled were local breed "Ankole", 67.0% (201/300) were crossbreeds and 5.3% (16/300) were Friesians. Samples were mainly collected from high throughput abattoirs (n = 280) compared to low throughput abattoirs (n = 20).

### Brucellosis seroprevalence

The seroprevalence of brucellosis in parallel was 20.7% (62/300) and 2.7% (8/300) using RBT and i-ELISA, respectively. The seroprevalence was 2.7%, (8/300) using both tests in series (Table 2).

**Table 2. Serological, bacterial culture, and PCR results of samples collected from slaughtered cattle in Rwanda.**

| Tested | RBT | i-ELISA | RBT&i-ELISA | Bacterial growth | ITS PCR | AMOS PCR | | | Bruce-ladder PCR | |
|---|---|---|---|---|---|---|---|---|---|---|
| N | n⁺ (%) | n⁺ (%) | n⁺ (%) | n⁺(%) | n⁺(%) | *B. abortus* | *B. melitensis* | *B. abortus&B. melitensis* | *B. abortus* | *B. melitensis* |
| 300 | 62 (20.7) | 8 (2.7) | 8 (2.7) | 87 (29.0) | 17 (5.7) | 4 | 3 | 4 | 5 | 6 |

RBT: Rose Bengal Test, i-ELISA: indirect enzyme-linked immunosorbent assay, ITS-PCR: the 16S-23S ribosomal interspacer region PCR, AMOS PCR: *B. abortus*, *B. melitensis*, *B. ovis*, and *B. suis* PCR.

### Brucellosis case status by bacterial culture and the ITS PCR assay

Of the tissues that were cultured onto the modified CITA medium, ITS PCR confirmed 5.7% (17/300) (Table 2, Fig 2). Therefore, the prevalence obtained by bacteriology, the gold standard, and confirmed by ITS PCR was 5.7% (17/300). The distribution of *Brucella* spp. was high in high throughput abattoirs compared to low throughput but without a significant difference and *Brucella* isolates were almost equally distributed in all five provinces. All isolates were from adult females. There was a significant difference (p = 0.02) between breeds with cross-breeds having *Brucella* infections compared to other breeds (Table 3).

### Differentiation of *Brucella* spp. by AMOS and Bruce-ladder PCR assays

The AMOS PCR identified *B. melitensis* and *B. abortus* (n = 3) mixed cultures, *B. abortus* (n = 3), and *B. melitensis* (n = 5) (Fig 3) from the 17 *Brucella* cultures (impure culture). The Bruce-ladder PCR assay identified *B. abortus* (n = 5), and *B. melitensis* (n = 6) (Fig 4).

The *Brucella* DNA detected by ITS, AMOS, and Bruce-ladder PCR assays (100.0%, 11/11) were from cattle that were either seropositive to RBT or i-ELISA. Of these 11 *Brucella* isolates, 10 were isolated from slaughtered cattle collected at high throughput abattoir. The 11 *Brucella* isolates that were identified in provinces are as follows: Eastern (n = 1), Kigali city (n = 2), Southern (n = 3), Western (n = 2), and Northern (n = 3). The 11 *Brucella* isolates stratified by breeds were Ankole (n = 1), crossbreds (n = 8), and Friesians (n = 2). There was no significant difference between the category of abattoirs, provinces, age, sex of animals, and the detection by ITS, AMOS, and Bruce-ladder PCR assays.

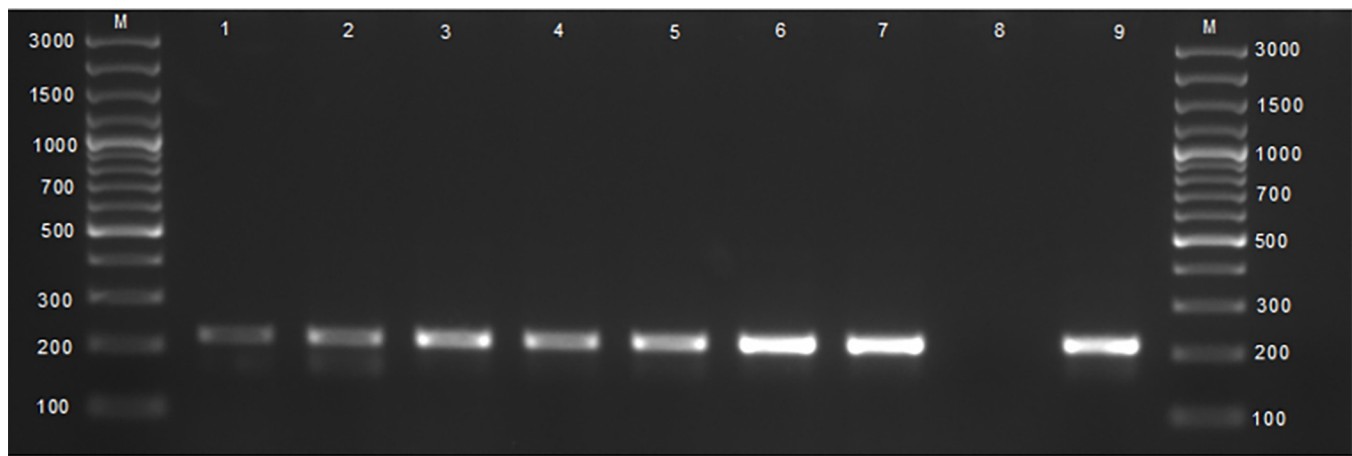

**Fig 2. Agarose gel electrophoresis of the 16-23S interspacer region (ITS) PCR products amplified from cultures of tissues from slaughtered cattle.** Lanes M: DNA Gene Ruler 100bp plus (Thermo Fischer Scientific, Johannesburg, South Africa), lanes 1–7: amplification of a 214 bp sequence of the genus *Brucella* spp., lane 8: negative control containing sterile water, lane 9: positive control with *B. abortus* RF544.

**Table 3. Univariable associations between animal characteristics and ITS PCR assay of *Brucella* spp. isolates from cultures of tissues of slaughtered cattle in Rwanda.**

| Variables | Categories | Tested | ITS PCR assay on culture isolates | | |
|---|---|---|---|---|---|
| | | | n⁺ (%) | Odds Ratio, 95% CI | *p*-value |
| Abattoir throughput | Low (ref) | 20 | 1 (5.00) | | |
| | High | 280 | 16 (5.71) | 1.15 (0.14, 9.15) | 0.894 |
| Provinces | Eastern (ref) | 70 | 2 (2.86) | | |
| | Kigali city | 30 | 3 (10.00) | 3.78 (0.60, 23.88) | 0.158 |
| | Northern | 50 | 4 (8.00) | 2.96 (0.52, 16.81) | 0.222 |
| | Southern | 80 | 3 (3.75) | 1.32 (0.21, 8.16) | 0.762 |
| | Western | 70 | 5 (7.14) | 2.62 (0.49, 13.96) | 0.261 |
| Age | Young (ref) | 31 | 0 (0.00) | | |
| | Adults | 269 | 17 (6.32) | 7.80 (0, Inf.) | 0.989 |
| Sex | Males (ref) | 13 | 0 (0.00) | | |
| | Females | 287 | 17 (5.92) | 2.68 (0, Inf.) | 0.989 |
| Breed | Ankole (ref) | 83 | 1 (1.20) | | |
| | Cross | 201 | 13 (6.47) | 5.67 (0.73, 44.06) | 0.0972 |
| | Friesian | 16 | 3 (18.75) | 18.92 (1.83, 195.97) | 0.0137 |

ref = reference level used in the analysis to compare infection; n+: number of positives; %: percentage, the 16S-23S ribosomal interspacer region (ITS), CI: confidence interval, inf. = infinity

Of the 5 variables that were considered in the univariable analysis, only breed met the criterion (p ≤ 0.20) for inclusion in the multivariable model.

## Discussion

This is the first report of *B. abortus* and *B. melitensis* isolated from cultures of cattle tissues collected from abattoirs. The overall seroprevalence obtained in this study among slaughtered cattle selected from all the thirty districts of Rwanda (2.9% for RBT and i-ELISA) was lower than the culture prevalence of 5.6% (17/300), which is the gold standard. The fact that the lower sensitivity culture method is higher than the seroprevalence is a clear indication that the confirmatory i-ELISA test must be validated for bovine in Rwanda as the cut-off values were determined in developed countries with low brucellosis prevalence and thus clearly underestimate the prevalence due to high cut-off values.

The overall seroprevalence of 2.9% was also lower than the rates reported in Rwanda in different studies that were conducted at farm level [16, 17, 37]. However, the seroprevalence obtained in this study is comparable with the rate (3.4%) reported at Gaoundere municipal abattoir in Cameroun using RBT and i-ELISA [38], and the 3.9% reported among slaughtered cattle in Nigeria [39], and the 5.5% reported among slaughtered cattle in Gauteng province, South Africa [40]. This suggests that the seroprevalence rates observed in abattoirs are usually

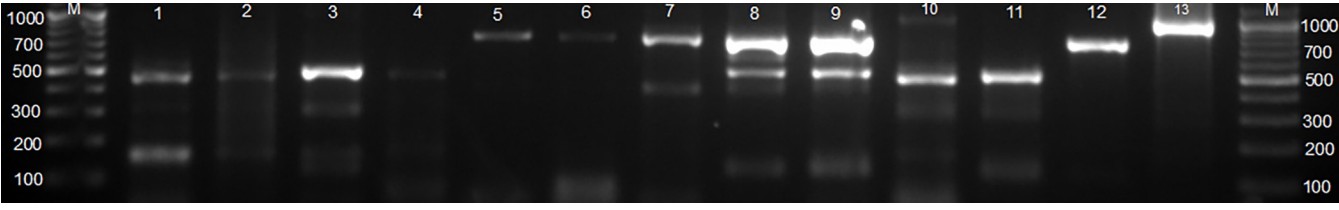

**Fig 3. Agarose gel electrophoresis for AMOS PCR products amplified from cultures of tissues from slaughtered cattle.** Lanes M: Gene Ruler 100 pb plus (ThermoFischer Scientific, Johannesburg, South Africa), lanes 1–4: *Brucella abortus* (496 bp), lanes 5–7: *B. melitensis* (731 bp), Lanes 9–10: mixed *B. melitensis* and *B. abortus*, lane 11: negative control containing sterile water, lane 12: positive control, *B. abortus* RF544, lane 13: positive control, *B. melitensis* rev 1.

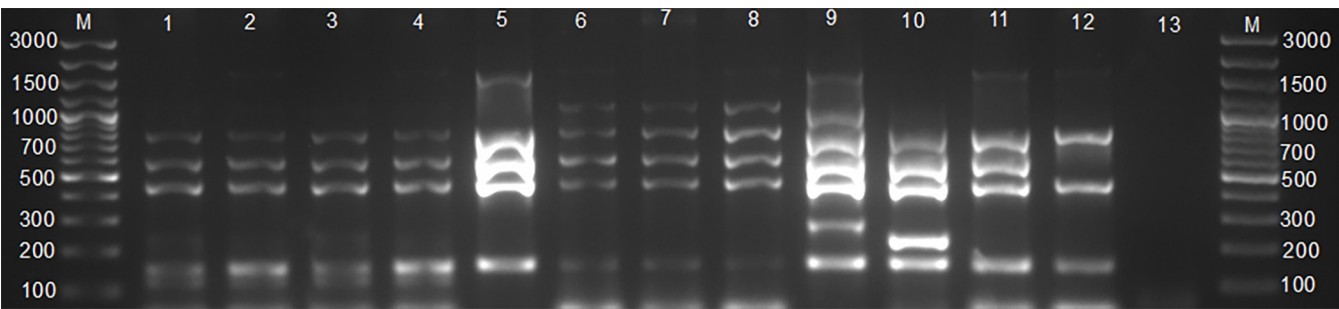

**Fig 4. Agarose gel electrophoresis for Bruce–ladder PCR products amplified from cultures of tissues from slaughtered cattle.** Lanes M: Gene Ruler 100 bp (ThermoFischer Scientific, Johannesburg, South Africa); lanes 1–5: *B. abortus*; lanes 6–8: *B. melitensis*; lane 9: positive control, *B. suis* ZW45, lane 10: positive control, *B. melitensis* rev 1, lane 11: *B. abortus* (REF 544), lane 12: positive control, *B. abortus* S 19, lane 13: negative control with sterile water.

lower compared to the seroprevalence recorded at the farm level which usually focuses on endemic zones while slaughtered cattle come from various locations (endemic and non-endemic zones).

Friesians were more likely to be seropositive in this study and was consistent with earlier studies in Pakistan where Holstein and Friesian cattle were more seropositive than indigenous breeds [41], and in Ethiopia [42]. This supports that exotic pure breeds like Friesians are more susceptible to brucellosis than crossbreeds and indigenous breeds [43] or were introduced in the herd with chronic infections with seronegative status but being chronically infected [12, 44]. When the acute brucellosis phase has passed, the infection stabilizes with the acquisition of herd immunity leading to fewer infectious discharges and non-visible symptoms [45].

The mixed infection caused by *B. abortus* and *B. melitensis* as well as the isolation of *B. melitensis* from slaughtered cattle indicate the cross-infection between both *Brucella* spp. and mixed farming of cattle and goats or sheep. The mixed infection and mixed farming were reported in our study that identified both pathogens in aborting goat flock in Rwanda [46]. The co-infection of *B. abortus* and *B. melitensis* has also been reported in slaughtered cattle in South Africa [40]. The isolation of *B. melitensis* in slaughtered cattle poses a risk to abattoir workers and consumers of contaminated milk and milk products as *B. melitensis* and *B. abortus* cause severe brucellosis in humans [47, 48]. There is a need for improvement in brucellosis control using vaccination as well as test-and-slaughter, coupled with raising awareness of all occupational groups as education was associated with a high awareness of brucellosis in Rwanda [17].

Both AMOS and Bruce-ladder PCR assays identified *B. abortus* and *B. melitensis* with the *B. abortus* being either biovars 1, 2, or 4 (identified by AMOS PCR) which will be identified in the future after purification of cultures using biotyping. In a previous study, *B. abortus* bv. 3 was identified in humans and animals in 1987 in Rwanda [20]. *B. abortus* bv. 3 and *B. melitensis* bv. 1 were reported in neighboring Uganda [49], Tanzania [50], Kenya [51], and South Africa [40]. Biotyping of *B. abortus* biovars is complex as characteristic typical for *B. abortus* bv .1, except $CO_2$ requirement for growth [52]. However, the *B. abortus* bv. 3 ref strain Tulya isolated from a human patient in Uganda grows in the absence of $CO_2$ and has been observed to occur within some biovars and changes with OIE biotyping profile [9, 50]. Hence classifying *B. abortus* bv. 3 strains should be carefully considered. Purifying and biotyping these cultures will be able to identify the biovar(s) and molecular characterization of the strains will allow trace-back studies. *Brucella abortus* and *B. melitensis* isolated in this study could originate from neighboring countries due to the repatriation of Rwandans and their livestock from Uganda and Tanzania as well as the importation of improved cattle breeds from various countries cannot be eliminated despite testing procedures [12].

## Conclusions

This study found the seroprevalence of brucellosis to be lower than the gold standard prevalence indicating that cut-off points of i-ELISA determined in Europe with brucellosis-free status or low prevalence, should be optimized for Rwanda as also reported by Mathew *et al*. (2015). This study identified *B. abortus* and *B. melitensis* as well as mixed infection in slaughtered cattle which is a result of the mixed livestock farming practice in Rwanda. These infections pose a risk of exposure potential to handlers of cattle, carcasses, and consumers of unpasteurized milk and milk products. Thus, vaccination and test-and-slaughter would significantly contribute to mitigating the disease. Furthermore, the introduction of an annual brucellosis-free certificate for large herds would contribute to mitigating brucellosis in the country.

## Supporting information

**S1 Data. Raw data in excel format and original gel images.**
(XLSX)

## Acknowledgments

The authors would like to acknowledge the National Reference Laboratory (NRL), Rwanda Agriculture and Animal Resources Development Board (RAB), the Department of Veterinary Services, and the University of Rwanda for the facilitation of this study. We also thank abattoir managers, inspectors and other abattoir workers, and laboratory staff at NRL and RAB for their assistance and good cooperation with the laboratory work.

## Author Contributions

**Conceptualization:** Jean Bosco Ntivuguruzwa, Henriette van Heerden.

**Data curation:** Jean Bosco Ntivuguruzwa.

**Formal analysis:** Jean Bosco Ntivuguruzwa.

**Funding acquisition:** Henriette van Heerden.

**Investigation:** Jean Bosco Ntivuguruzwa.

**Methodology:** Jean Bosco Ntivuguruzwa, Henriette van Heerden.

**Project administration:** Henriette van Heerden.

**Software:** Jean Bosco Ntivuguruzwa.

**Supervision:** Francis Babaman Kolo, Henriette van Heerden.

**Validation:** Henriette van Heerden.

**Writing – original draft:** Jean Bosco Ntivuguruzwa.

**Writing – review & editing:** Jean Bosco Ntivuguruzwa, Francis Babaman Kolo, Emil Ivan Mwikarago, Henriette van Heerden.

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
