## [Decision Letter · Decision Letter 0]

5 Jul 2022

PONE-D-21-38427Seroprevalence of brucellosis and molecular characterization of Brucella spp. from slaughtered cattle in Rwanda

PLOS ONE

Dear Dr. Ntivuguruzwa,

Thank you for submitting your manuscript to PLOS ONE. After careful consideration, we feel that it has merit but does not fully meet PLOS ONE’s publication criteria as it currently stands. Therefore, we invite you to submit a revised version of the manuscript that addresses the points raised during the review process.

In addition to the comments provided by the reviewers please address the following:

Line 26-27: parallel interpretation means positive in at least one test. So, what the authors mentioned is not parallel but the result of serial interpretation i.e, positive in both tests.

Line 56: It was not clear what the authors wanted to mean by “As brucellosis is a herd disease”

Line 105: 0.5% should be5% ; what was the basis to consider a 5% expected prevalence? "The expected prevalence was estimated at 0.5%." Here the word "estimated" is not correct.

Line 200: Brucellosis seroprevalence among slaughter cattle in Rwanda; this subsection can be replaced by “Descriptive statistics” and lines 201 to 205 could be placed under this subsection.

Then the remaining lines can be placed under another sub-section like Brucellosis seroprevalence.

Please insert another table [preferably Table 2] summarizing all test results

Please define brucellosis case status [preferably if culture or PCR positive] and only show the distribution of the brucellosis according to the different variables presented in Table 2 deleting the distribution of brucellosis based on RBT and i-ELISA [now it will become Table 3]. Please recheck the odds ratio as also suggested by one reviewer.

Finally, please evaluate the performance of RBT and i-ELISA considering culture/PCR the as gold standard.

We look forward to receiving your revised manuscript.

Kind regards,

A. K. M. Anisur Rahman, Ph.D.

Academic Editor

PLOS ONE

https://journals.plos.org/plosone/s/file?id=ba62/PLOSOne_formatting_sample_title_authors_affiliations.pdf".

 “HvH received funds from the Institute of Tropical Medicine Antwerp (https://www.itg.be)

in collaboration with the Department of Veterinary Tropical Diseases (https://www.up.ac.za/veterinary-tropical-diseases). “ 

“The authors would like to acknowledge the Institute of tropical medicine, Belgium, and the Department of Veterinary Tropical Disease, South Africa, for funding the project. Our acknowledgments also go to the National Reference Laboratory, Department of Veterinary Services, and the University of Rwanda for the facilitation of this study. We also thank abattoir managers, inspectors and other abattoir workers, laboratory staff at NRL, and RAB for their assistance and good cooperation.”

“HvH received funds from the Institute of Tropical Medicine Antwerp (https://www.itg.be)

in collaboration with the Department of Veterinary Tropical Diseases (https://www.up.ac.za/veterinary-tropical-diseases).”

The authors declare that there are no conflicts of interest. The funders had no role in the design of the study; in the collection, analyses, or interpretation of data, in the writing of the manuscript, or in the decision to publish the results.”

6. We noted in your submission details that a portion of your manuscript may have been presented or published elsewhere. [Figure 1 was was submitted in PLOS Neglected diseases in a different manuscript which is still under consideration. The figure describes the study area which was the same for both manuscripts (this one is on brucellosis while the other is on bovine tuberculosis) but samples and data are different] Please clarify whether this [conference proceeding or publication] was peer-reviewed and formally published. If this work was previously peer-reviewed and published, in the cover letter please provide the reason that this work does not constitute dual publication and should be included in the current manuscript.

7. PLOS ONE now requires that authors provide the original uncropped and unadjusted images underlying all blot or gel results reported in a submission’s figures or Supporting Information files. This policy and the journal’s other requirements for blot/gel reporting and figure preparation are described in detail at https://journals.plos.org/plosone/s/figures#loc-blot-and-gel-reporting-requirements and https://journals.plos.org/plosone/s/figures#loc-preparing-figures-from-image-files. When you submit your revised manuscript, please ensure that your figures adhere fully to these guidelines and provide the original underlying images for all blot or gel data reported in your submission. See the following link for instructions on providing the original image data: https://journals.plos.org/plosone/s/figures#loc-original-images-for-blots-and-gels.

8. Your ethics statement should only appear in the Methods section of your manuscript. If your ethics statement is written in any section besides the Methods, please delete it from any other section.

9. We note that [Figure 1] in your submission contain [map/satellite] images which may be copyrighted. All PLOS content is published under the Creative Commons Attribution License (CC BY 4.0), which means that the manuscript, images, and Supporting Information files will be freely available online, and any third party is permitted to access, download, copy, distribute, and use these materials in any way, even commercially, with proper attribution. For these reasons, we cannot publish previously copyrighted maps or satellite images created using proprietary data, such as Google software (Google Maps, Street View, and Earth). For more information, see our copyright guidelines: http://journals.plos.org/plosone/s/licenses-and-copyright.

10. Please include captions for your Supporting Information files at the end of your manuscript, and update any in-text citations to match accordingly. Please see our Supporting Information guidelines for more information: http://journals.plos.org/plosone/s/supporting-information.

Reviewers' comments:

Reviewer's Responses to Questions

**Comments to the Author**

1. Is the manuscript technically sound, and do the data support the conclusions?

Reviewer #1: Yes

Reviewer #2: Yes

2. Has the statistical analysis been performed appropriately and rigorously? 

Reviewer #1: No

Reviewer #2: Yes

3. Have the authors made all data underlying the findings in their manuscript fully available?

Reviewer #1: Yes

Reviewer #2: Yes

4. Is the manuscript presented in an intelligible fashion and written in standard English?

Reviewer #1: Yes

Reviewer #2: Yes

5. Review Comments to the Author

Reviewer #1: Dear Authors,

This is an interesting piece of work that will have an important implication on policy in Rwanda. I must congratulate the team for a well-conducted and written manuscript. However please double check odds ratio and its confidence interval which seems incomplete.

All the best for your future.

“Thank You”

Reviewer #2: Seroprevalence of brucellosis and characterisation of Brucella spp. from slaughtered cattle in Rwanda is well written paper about a very important zoonotic disease of human and animal health

Following are suggestions for improvement of paper before accepted for publication

line 36-37. is test and slaughter policy applicable in Rwanda

line 62-63. remove underline sign from species name of Brucella

line 148, have you performed microscopy or biochemical testing for Brucella susceptable isolates

line 149, there is a need to add details of positive and negative controls used in genus specific PCRs

line 179-180. what awas product length of B. abortus and melitensis in case of AMOS and Bru-ladder PCR assays

-it will be a good addition, if you can add sequencing results too (if possible, otherwise you can try it next time)

6. PLOS authors have the option to publish the peer review history of their article (what does this mean?). If published, this will include your full peer review and any attached files.

Reviewer #1: No

Reviewer #2: No

---

## [Author Response · Author response to Decision Letter 0]

12 Aug 2022

Response to reviewers

July 30th, 2022

Editor, Reviewer 1 and Reviewer 2 

PLOS One 

Re: revision of the manuscript [PONE-D-21-38427]-[EMID: 82b0bed3761197ba]

 Dear Editor and reviewers,

The authors would like to acknowledge your valuable inputs. We thank you for reviewing our manuscript entitled “Seroprevalence of brucellosis and molecular characterization of Brucella spp. from slaughtered cattle in Rwanda”. Your comments were very clear and useful to improve the quality of this manuscript. We have revised the manuscript considering all issues mentioned in your comments. Outlined below are the point-by-point responses. 

Editor 

Comment: line 26-27: parallel interpretation means positive in at least one test. So, what the authors mentioned is not parallel but the result of serial interpretation i.e, positive in both tests.

Response: the comment was addressed. Parallel replaced by series 

Comment: line 56: It was not clear what the authors wanted to mean by “As brucellosis is a herd disease”

Response: the sentence was rephrased as follows: “to detect brucellosis at herd level, the ……” 

Comment: line 105: 0.5% should be 5%; what was the basis to consider a 5% expected prevalence? "The expected prevalence was estimated at 0.5%." Here the word "estimated" is not correct.

Response: the comment was addressed. 0.5% replaced by 5% and the sentence was rephrased as follows: “the expected prevalence was 5% based on previous studies (Ntivuguruzwa et al., 2020)”. 

Comment: line 200: Brucellosis seroprevalence among slaughter cattle in Rwanda; this subsection can be replaced by “Descriptive statistics” and lines 201 to 205 could be placed under this subsection.

Then the remaining lines can be placed under another sub-section like Brucellosis seroprevalence.

Response: the comment was addressed. A subsection of descriptive statistics was added 

Comment: please insert another table [preferably Table 2] summarizing all test results

Response: a separate table 2 was added as follows: 

Table 2. Serological, bacterial culture and PCR results of samples collected from slaughtered cattle in Rwanda 

Tested RBT i-ELISA RBT&i-ELISA Culture ITS PCR AMOS PCR Bruce-ladder PCR 

N n+ (%) n+ (%) n+ (%) n+(%) n+(%) B. abortus B. melitensis B. abortus&B. melitensis B. abortus B. melitensis 

300 62 (20.7) 8 (2.7) 8 (2.7) 87 (29.0) 17 (5.7) 4 3 4 5 6

Comment: Please define brucellosis case status [preferably if culture or PCR positive] and only show the distribution of the brucellosis according to the different variables presented in Table 2 deleting the distribution of brucellosis based on RBT and i-ELISA [now it will become Table 3]. 

Response: table was readjusted as follows:

Variables Categories Tested ITS PCR assay on culture isolates

 n+ (%) Odds Ratio p-value

Abattoirs High throughput 280 16 (5.71) 0.02-6.22 1

 Low throughput 20 1 (5.00) 

Provinces Eastern 70 2 (2.86) Undetermined 0.43

 Kigali city 30 3 (10.00) 

 Northern 50 4 (8.00) 

 Southern 80 3 (3.75) 

 Western 70 5 (7.14) 

Age Adults 269 17 (6.32) 0.00-2.09 0.23

 Young 31 0 (0.00) 

Sex Females 287 17 (5.92) 0.00-5.75 1

 Males 13 0 (0.00) 

Breed Ankole 83 1 (1.20) Undetermined 0.02

 Cross 201 13 (6.47) 

 Friesian 16 3 (18.75) 

Comment: Please recheck the odds ratio as also suggested by one reviewer.

Response: the odds ration were checked and corrected 

Comment: Finally, please evaluate the performance of RBT and i-ELISA considering culture/PCR the as gold standard.

Response: the comment was addressed as follows: 

“The performance of RBT and i-ELISA indicated by the seroprevalence rate (2.7%) was low compared to the prevalence obtained by culture and ITS PCR (5.7%)”. 

Review Comments to the Author

Reviewer #1: Dear Authors, 

This is an interesting piece of work that will have an important implication on policy in Rwanda. I must congratulate the team for a well-conducted and written manuscript. However please double check odds ratio and its confidence interval which seems incomplete. All the best for your future. 

“Thank You”

Response: Thank you so much. The odds ratio were checked and corrected. 

Reviewer #2: Seroprevalence of brucellosis and characterisation of Brucella spp. from slaughtered cattle in Rwanda is well written paper about a very important zoonotic disease of human and animal health. Following are suggestions for improvement of paper before accepted for publication 

Response: Thank you so much

Comment: Line 36-37. is test and slaughter policy applicable in Rwanda 

Response: the comment was addressed as follows: 

It is essential to urgently strengthen a coordinated national bovine brucellosis vaccination and initiate test-and-slaughter program which is not presently applicable in Rwanda. 

Comments: Line 62-63. remove underline sign from species name of Brucella

Response: the underline sign was removed: PCR assays which differentiate B. abortus bv.1, 2, 4, B. melitensis bv.1, 2, 3, B. ovis, and B. suis bv.1 (AMOS PCR) in 24 hours from cultures

Comment: line 148, have you performed microscopy or biochemical testing for Brucella susceptable isolates 

Response: we did not perform microscopy nor biochemical testing. All suspect cultures were screened with ITS PCR 

Comment: Line 149, there is a need to add details of positive and negative controls used in genus specific PCRs 

Response: the details for the positive and negative controls were indicated in line 153 as follows: “with the B. abortus RF 544 (Onderstepoort Biological Products, South Africa) as a positive control. For the negative control, we used ultrapure sterile water (Onderstepoort Biological Products, South Africa)”. 

Comment: Line 179-180. what a was product length of B. abortus and melitensis in case of AMOS and Bru-ladder PCR assays-it will be a good addition, if you can add sequencing results too (if possible, otherwise you can try it next time)

Response: The expected product sizes in AMOS PCR assay are 498 bp and 731 bp for B. abortus and B. melitensis, respectively. The expected product sizes for B. abortus in Bruce-ladder PCR assay are 152 bp, 498 bp, 587 bp, 587 bp, and 794 bp; they are 152 bp, 498 bp, 587 bp, 794 bp and 1071 bp for B. melitensis.

---

## [Decision Letter · Decision Letter 1]

12 Sep 2022

PONE-D-21-38427R1

Seroprevalence of brucellosis and molecular characterization of Brucella spp. from slaughtered cattle in Rwanda

PLOS ONE

Dear Dr. Ntivuguruzwa,

Thank you for submitting your manuscript to PLOS ONE. After careful consideration, we feel that it has merit but does not fully meet PLOS ONE’s publication criteria as it currently stands. Therefore, we invite you to submit a revised version of the manuscript that addresses the points raised during the review process.

ACADEMIC EDITOR:

Please ignore my previous comment regarding the evaluation of RBT and i-ELISA considering culture/PCR as gold-standard as it was not performed properly and also it may not be possible to evaluate efficiently due small size of culture /PCR positive samples.Please estimate odds ratio appropriately. There should have one reference for every variables and usually the category with lowest prevalence is considered as reference. For example, Low throughput for abattoirs, eastern for provinces, etc.Please replace "univariate" with "univariable" from Table 3 title.Other than reference category, every category of a variable should have an odds ratio. What did you mean by "undetermined"?Please add before line 177: The brucellosis case status was defined based on the results of culture and ITS PCR tests in series.Lines 177-180: Please add the following: The chi-square test was used to evaluate the univariable association between the explanatory variables and brucellosis case status. Any explanatory variable associated with brucellosis at a P≤0.20 was included in the multivariable logistic regression analysis to identify risk factors.Lines 222-223: The brucellosis case status was considered as prevalence based on bacterial growth and  confirmation by ITS PCR. Please delete this sentence.Please delete lines 249-250Please delete lines 334-346 as these results are not statistically significant.The authors could add a discussion about the 70 samples which were culture positive but PCR negativePlease elaborate ITS and AMOS in the Tables 2 and 3 foot noteLine 348: Please replace "rate" with "prevalence"

We look forward to receiving your revised manuscript.

Kind regards,

A. K. M. Anisur Rahman, Ph.D.

Academic Editor

PLOS ONE

Journal Requirements:

Reviewers' comments:

Reviewer's Responses to Questions

**Comments to the Author**

1. If the authors have adequately addressed your comments raised in a previous round of review and you feel that this manuscript is now acceptable for publication, you may indicate that here to bypass the “Comments to the Author” section, enter your conflict of interest statement in the “Confidential to Editor” section, and submit your "Accept" recommendation.

Reviewer #1: All comments have been addressed

Reviewer #2: All comments have been addressed

2. Is the manuscript technically sound, and do the data support the conclusions?

Reviewer #1: Yes

Reviewer #2: Yes

3. Has the statistical analysis been performed appropriately and rigorously? 

Reviewer #1: Yes

Reviewer #2: Yes

4. Have the authors made all data underlying the findings in their manuscript fully available?

Reviewer #1: Yes

Reviewer #2: Yes

5. Is the manuscript presented in an intelligible fashion and written in standard English?

Reviewer #1: Yes

Reviewer #2: Yes

6. Review Comments to the Author

Reviewer #1: The authors have addressed all the comments as suggested and it is a good piece of work which will be useful in the policy implications in Rwanda. Congratulations to the team for the great work.

Thanks and Regards

Reviewer #2: Authors have addressed all of my comments adequately

Hopefully this paper will be a nice piece of research work for brucellosis researchers across the globe.

7. PLOS authors have the option to publish the peer review history of their article (what does this mean?). If published, this will include your full peer review and any attached files.

Reviewer #1: **Yes: **Dr Ritik Agrawal

Reviewer #2: No

---

## [Author Response · Author response to Decision Letter 1]

17 Oct 2022

Response to reviewers

October 17th, 2022

Editor,

PLOS One 

Re: revision of the manuscript [PONE-D-21-38427]-[EMID: 82b0bed3761197ba]

 Dear Editor 

The authors would like to acknowledge your valuable input. We thank you for reviewing our manuscript entitled “Seroprevalence of brucellosis and molecular characterization of Brucella spp. from slaughtered cattle in Rwanda”. Your comments were very clear and useful to improve the quality of this manuscript. We have revised the manuscript considering all issues mentioned in your comments. Outlined below are the point-by-point responses. 

Comment: 

1. Please ignore my previous comment regarding the evaluation of RBT and i-ELISA considering culture/PCR as gold-standard as it was not performed properly and also it may not be possible to evaluate efficiently due small size of culture /PCR positive samples.

Response: the sentence was deleted 

Comment: 

2. Please estimate odds ratio appropriately. There should have one reference for every variables and usually the category with lowest prevalence is considered as reference. For example, Low throughput for abattoirs, eastern for provinces, etc.

Response: the univariable table was revised to remove the odds ratio. The variables with P ≤ 0.20 were selected for the multivariable analysis table. 

Table 4. Results of the multivariable logistic regression analysis between animal characteristics and ITS PCR assay of Brucella spp. isolates from cultures of tissues of slaughtered cattle in Rwanda

Variables Category Odds Ratios 95% CI p-Value

Breeds Ankole a 

 Crossbreed 5.7 0.73-44.06 0.097

 Friesian 18.9 1.83-195.96 0.014

Comment:

3. Please replace "univariate" with "univariable" from Table 3 title.

Response: corrected 

Comment: 

4. Other than reference category, every category of a variable should have an odds ratio. What did you mean by "undetermined"?

Response: 

Comment:

5. Please add before line 177: The brucellosis case status was defined based on the results of culture and ITS PCR tests in series.

Response: corrected 

Comment: 

6. Lines 177-180: Please add the following: The chi-square test was used to evaluate the univariable association between the explanatory variables and brucellosis case status. Any explanatory variable associated with brucellosis at a P≤0.20 was included in the multivariable logistic regression analysis to identify risk factors.

Response: sentences were added. 

Comment: 

7. Lines 222-223: The brucellosis case status was considered as prevalence based on bacterial growth and confirmation by ITS PCR. Please delete this sentence.

Response: the sentence was deleted 

Comment: 

8. Please delete lines 249-250

Response: corrected 

Comment: 

9. Please delete lines 334-346 as these results are not statistically significant.

Response: corrected 

Comment: 

10. The authors could add a discussion about the 70 samples which were culture positive but PCR negative

Response: All samples that presented growth on the CITA agar plate were subjected to ITS PCR for confirmation. The 70 samples were not culture-positive for Brucella and therefore can not be discussed. 

Comment: 

11. Please elaborate ITS and AMOS in the Tables 2 and 3 foot note

Response: corrected 

Comment: 

12. Line 348: Please replace "rate" with "prevalence"

Response: corrected

---

## [Editor Report · Decision Letter 2]

20 Oct 2022

PONE-D-21-38427R2Seroprevalence of brucellosis and molecular characterization of Brucella spp. from slaughtered cattle in RwandaPLOS ONE

Dear Dr. Ntivuguruzwa,

Thank you for submitting your manuscript to PLOS ONE. After careful consideration, we feel that it has merit but does not fully meet PLOS ONE’s publication criteria as it currently stands. Therefore, we invite you to submit a revised version of the manuscript that addresses the points raised during the review process.

ACADEMIC EDITOR:I still need clarification about one of my previous comment: The authors could add a discussion about the 70 samples which were culture positive but PCR negative

Response: All samples that presented growth on the CITA agar plate were subjected

to ITS PCR for confirmation. The 70 samples were not culture-positive for Brucella and

therefore can not be discussed.If this is the case then please clarify the figure 89 (29%) under the column 'culture'.Please remove Table 4 as it is not based on the results of a multivariable model. Multivariable model must contain at least two variables. As only one variable is associated with a p value <0.20 in the univariable screening there is no need to run the multivariable model. Please include odds ratio and their 95% Confidence interval in the Table 3. ==============================

Kind regards,

A. K. M. Anisur Rahman, Ph.D.

Academic Editor

PLOS ONE
---

## [Author Response · Author response to Decision Letter 2]

3 Nov 2022

Response to reviewers

October 29th, 2022

Editor,

PLOS One 

Re: revision of the manuscript [PONE-D-21-38427]-[EMID: 82b0bed3761197ba]

 Dear Editor 

The authors would like to acknowledge your valuable input. We thank you for reviewing our manuscript entitled “Seroprevalence of brucellosis and molecular characterization of Brucella spp. from slaughtered cattle in Rwanda”. Your comments were very clear and useful to improve the quality of this manuscript. We have revised the manuscript considering all issues mentioned in your comments. Outlined below are the point-by-point responses. 

Comment: 

I still need clarification about one of my previous comment: The authors could add a discussion about the 70 samples which were culture positive but PCR negative

Response: All samples that presented growth on the CITA agar plate were subjected to ITS PCR for confirmation. The 70 samples were not culture-positive for Brucella and therefore cannot be discussed.

If this is the case then please clarify the figure 89 (29%) under the column 'culture'.

Response: 

CITA medium and most other selective media focus on isolating Brucella but each medium still does not allow exclusive growth of Brucella. They reduce most contaminant bacteria but some others still grow. Thus other contaminants will grow on CITA and other selective media (Ledwaba et al., 2020). Considering the growth of contaminant bacteria that can even hide the Brucella growth, we screened each growth with ITS PCR for confirmation.

Figure 87 (29%) under column “culture” indicates bacterial growth including contaminants. Therefore, “culture” was changed to “bacterial growth” to avoid confusion. 

Comment: 

Please remove Table 4 as it is not based on the results of a multivariable model. A Multivariable model must contain at least two variables. As only one variable is associated with a p value <0.20 in the univariable screening there is no need to run the multivariable model. Please include odds ratio and their 95% Confidence interval in the Table 3.

Response: 

The table was deleted. 

Odds ratio and their 95% confidence interval were included in the univariable table. 

Variables Categories Tested ITS PCR assay on culture isolates

 n+ (%) Odds Ratio, 95% CI p-value

Abattoir throughput Low (ref) 20 1 (5.00) 

 High 280 16 (5.71) 1.15 (0.14, 9.15) 0.894

Provinces Eastern (ref) 70 2 (2.86) 

 Kigali city 30 3 (10.00) 3.78 (0.60, 23.88) 0.158

 Northern 50 4 (8.00) 2.96 (0.52, 16.81) 0.222

 Southern 80 3 (3.75) 1.32 (0.21, 8.16) 0.762

 Western 70 5 (7.14) 2.62 (0.49, 13.96) 0.261

Age Young (ref) 31 0 (0.00) 

 Adults 269 17 (6.32) 7.80 (0, Inf.) 0.989

Sex Males (ref) 13 0 (0.00) 

 Females 287 17 (5.92) 2.68 (0, Inf.) 0.989

Breed Ankole (ref) 83 1 (1.20) 

 Cross 201 13 (6.47) 5.67 (0.73, 44.06) 0.0972

 Friesian 16 3 (18.75) 18.92 (1.83, 195.97) 0.0137

ref=reference level used in the analysis to compare infection; n+: number of positives; %: percentage, the 16S-23S ribosomal interspacer region (ITS), CI: confidence interval, inf.=infinity 

Kind regards

---

## [Editor Report · Decision Letter 3]

6 Nov 2022

Seroprevalence of brucellosis and molecular characterization of Brucella spp. from slaughtered cattle in Rwanda

PONE-D-21-38427R3

Dear Dr. Ntivuguruzwa,

We’re pleased to inform you that your manuscript has been judged scientifically suitable for publication and will be formally accepted for publication once it meets all outstanding technical requirements.

Kind regards,

A. K. M. Anisur Rahman, Ph.D.

Academic Editor

PLOS ONE
---

## [Editor Report · Acceptance letter]

10 Nov 2022

PONE-D-21-38427R3 

Seroprevalence of brucellosis and molecular characterization of *Brucella* spp. from slaughtered cattle in Rwanda 

Dear Dr. Ntivuguruzwa:

I'm pleased to inform you that your manuscript has been deemed suitable for publication in PLOS ONE. Congratulations! Your manuscript is now with our production department. 

Kind regards, 

on behalf of

Dr. A. K. M. Anisur Rahman 

Academic Editor

PLOS ONE